# *"I was screaming hallelujah"*: Patient and provider perceptions of blood-based testing for colorectal cancer screening

**Jennifer L. Schneider**⬤*, **Cheryl A. Johnson, Charisma Jenkins, Rajasekhara Mummadi, Gloria D. Coronado**

Center for Health Research, Kaiser Permanente, Portland, Oregon, United States of America

* jennifer.l.schneider@kpchr.org

## Abstract

### Background

Blood-based tests for colorectal cancer (CRC) screening can offer many advantages over stool-based tests such as FIT. Yet, we know little about patients' and providers' perceptions of this type of test. We report findings from a qualitative study comparing patient and provider perceptions of blood-based testing for CRC screening.

### Methods

Patient participants were aged 45–75 years and members of a large, integrated health system. Participants were mailed, but did not complete, a FIT through an organized FIT-screening program and were scheduled for a health-care encounter at any of nine clinical sites. Participants were consented to complete a blood draw. We used purposive sampling to select and recruit patients (who did and did not complete the blood test) and providers/specialists who would be involved in offering the blood test to patients or explaining results. We administered telephone interviews using a semi-structured interview guide and recorded and transcribed all interviews, then coded and analyzed content.

### Results

We interviewed 15 patients (11 completed and 4 did not complete the blood test) and 5 providers (3 primary care providers, one gastroenterologist (GI), and one GI medical assistant). Patients were enthusiastic about completing a blood test, citing the simplicity, ease, convenience, and high perceived accuracy of the test. Providers were also receptive to a blood-based option, if adequate test performance could be achieved and if they have information that informs patients about the pros and cons of blood-based screening versus other screening tests.

### Conclusions

Patients and providers were willing and enthusiastic about blood-based CRC screening tests. Future research focusing on performance and communication is needed.

**Data Availability Statement:** All relevant data are within the paper and its Supporting Information files.

**Funding:** This study was funded by Guardant Health (Home - GuardantHealth) through a contract to the Kaiser Permanente Center for Health Research. The funders processed blood samples for the study, but otherwise had no role in the study design, data collection, decision to publish or preparation of the manuscript.

**Competing interests:** I have read the journal's policy and the authors of this manuscript have the following competing interests: From 2020- 2022, Dr. Coronado served as a scientific advisor for Exact Sciences, through a contract with the Kaiser Permanente Center for Health Research. This does not alter our adherence to PLOS ONE policies on sharing data and materials.

## Introduction

Patients have multiple options for colorectal cancer (CRC) screening, ranging from most invasive (colonoscopy or flexible sigmoidoscopy) to least invasive (stool testing. i.e., fecal immunochemical test or FIT) [1]. Blood testing for cell-free DNA shed by tumors is the newest option [2]. However, few studies have assessed acceptability of the test among patients and providers. Our findings address this evidence gap.

Blood-based tests for CRC can offer advantages over stool-based tests. Blood draws are usually convenient, require no preparation, and many patients undergo them routinely during point of care visits. Moreover, discomfort with handling fecal matter is a known deterrent to stool-testing options [3], raising the possibility that blood-based testing could be more widely acceptable and result in greater adherence to screening recommendations. However, previous literature on patients' and providers' perceptions of blood-based testing for CRC screening is scarce. Davis and colleagues recently conducted a scoping review that included six studies focused on blood testing for CRC screening in which patients favored blood-based testing based on non-invasiveness, perceived safety, convenience, and effectiveness. No studies reported on primary care provider perceptions of testing [4].

As part of a larger pilot study of blood-test adherence conducted in an integrated care setting, we examined perceptions of blood-based CRC screening among providers and patients who had declined to participate in FIT testing in the past, and were offered blood-based testing opportunistically as part of standard health encounters. The test used in this study was Shield (GuardantHealth, Redwood City, CA). Shield uses next-generation sequencing to detect colorectal neoplasia-derived tumor signals in blood by analyzing cell-free DNA for genomic or epigenomic alternations associated with CRC. The test provides a binary result (normal vs. abnormal); abnormal test results should be followed up with a colonoscopy. Recent findings from a study involving more than 20,000 average-risk patients demonstrated an 83% sensitivity for CRC detection, a 13% sensitivity for advanced adenoma detection, and a 90% specificity for the absence of CRC or advanced adenoma [5–7], surpassing the Centers for Medicare & Medicaid performance thresholds (sensitivity ≥74% and specificity ≥90%, which is the performance of FIT) [8]. Notably, this level of detection for advanced adenomas is poorer than for FIT (23–28% sensitivity, 94–96% specificity) and FIT-DNA (43% sensitivity; 89% specificity) [1]. The test is commercially available, has not received FDA approval (as of this writing), and has a manufacturer-recommended frequency of every three years for average-risk adults, which is commercially available but not approved by the US Food and Drug Administration at the time of this project [5]. Our study is one of the first to look at acceptability of a blood-based test to patients and providers, thus filling an important research gap. Our findings can help guide potential future implementation of blood-based CRC screening in primary care settings.

## Methods

### Study setting and background

The parent study took place at Kaiser Permanente Northwest (KPNW), an integrated health system with clinics in Oregon and Southwest Washington serving a primarily urban, metropolitan area, and assessed adherence to an offer of blood-based screening for CRC. The study identified and recruited health plan members who, in the prior 3 to 9 months, were given a FIT test as part of standard practice but never returned the FIT, classifying them as individuals who chose not to actively pursue the FIT test option (passive decliners). Eligible health plan members for the parent study had an upcoming visit scheduled at one of nine clinics (located

near two labs where blood draws were performed). During their scheduled visit, individuals were offered the option of completing the blood-based CRC screening test and if interested they completed written informed consent prior to the blood draw. During consent for the blood draw, participating individuals were also informed they may be contacted for an interview to learn about their experiences completing (or not) the blood-based CRC screening option.

## Data collection and recruitment

The qualitative team (JLS, CAJ) developed semi-structured interview guides (see S1 File) for both patient and provider interviews to better understand acceptability of a blood-based CRC screening option. Guides were generally informed by prior literature and reviewed by the larger research team (JLS, CAJ, CJ, RM, and GDC), all of whom have experience with and expertise in CRC screening options. Questions in the patient guide asked about prior CRC screening history and experiences, including barriers to prior FIT completion; motivations for completing the blood-test or not; general experience of the blood-test draw/appointment; experience receiving results from the blood-test; and future actions and possible improvement. For providers, we explored prior awareness of blood-based CRC screening options, reactions to the blood test offered in the study, including advantages and disadvantages; and recommendations for how to potentially incorporate the blood-based test into current CRC screening approaches (e.g., mailed FIT). Patient and provider interviews were conducted by phone by trained and experienced qualitative staff (JLS, CAJ), lasted about 45 to 60 minutes, and occurred between January and March 2023. Patient interviews occurred approximately 4 to 8 weeks after the completed or booked blood draw appointment. For the provider interviews, we also emailed them a visual (see S2 File) of what was known about the performance, interval, and cost of the blood test to-date to aid in the interview discussion. The visual showed test reliability of 91% sensitivity and 94% specificity for CRC, and a 20% sensitivity to detect advanced adenomas, based on preliminary reports (more robust estimates based on larger sample sizes are provided in the methods, above) [9]. All interview materials and processes were reviewed and approved by the Institutional Review Board at KPNW.

## Patients

We used purposeful sampling [10, 11] to select patients who were offered and either did or did not complete the blood-based CRC screening test in the main study. Our goal was to explore if the motivations and/or barriers to completing the blood-test option were different or similar across those patients that agreed to and completed or did not complete the blood test. We used a tracking tool from the parent study to generate a list of possible interviewees. Patients who were about 4 to 8 weeks out from their completed blood draw or booked/ uncompleted blood draw were sent an introduction letter, followed by up to four phone outreach attempts by qualitative staff (JLS, CAJ). Patients were offered a $25 gift card for completing the interview. Given the pilot-nature of the parent study, our overall goal was about 12 to 15 interviews, and we attempted to interview patients representing a range of genders, ages, and completion status. We intentionally recruited more patients who completed the blood test, as one of our primary aims was to understand the actual experience and acceptability of choosing the blood test after previously not completing the health system's standard screening option, a FIT. We did not attempt to recruit participants from a range of income quartiles, education backgrounds, or who had never actively been offered a FIT in the past.

### Providers

We used purposeful snowball sampling [10, 12] to identify primary care and gastroenterology providers involved in offering or discussing CRC screening with patients, and with past or current leadership roles pertaining to quality metrics and population-based screening. We first identified and interviewed the gastroenterologist involved with our parent study and asked this provider for recommendations for other GI staff or PCPs who might want to participate. PCPs did not necessarily have patients in the study. Additionally, after each interview we asked for recommendations of providers in leadership roles who might have experience and opinions in the topic area. Providers were recruited by email, and no incentive was offered per clinical guidelines.

### Analysis

All interviews were audio-recorded and transcribed for analysis. A professional transcription service was utilized to aid in accuracy of transcripts. Additionally, prior to engaging in formal analysis, qualitative staff (JLS) reviewed all transcripts for quality and accuracy, including checking transcription content against the audio-recording as needed. Given the small number of interviews, our analysis goal was not thematic saturation of the interview data, but rather exploration of barriers to and facilitators of the acceptability of the blood-based CRC screening option from both patient and provider perspectives. Overall, the qualitative team (JLS, CAJ) followed an iterative content analysis approach using topical coding techniques [13–16]. First, they reviewed a sub-sample of patient and provider interview transcripts to aid in development of a code template within Microsoft Word. Next, the team (JLS, CAJ) reviewed the transcripts multiple times, and extracted relevant text from the transcripts by cutting and pasting it into the related content areas of the code template. The code templates reflected topical areas from the interview guides and emergent content from the reviewed transcripts (see S3 File for code template examples).

From this process, summarization reports of topically coded text were developed for both the provider and patient interviews [17]. The summarization reports were re-reviewed and refined by the qualitative team to identify key content findings and representative quotes for patients and providers. These summary reports with illustrative quotes were then reviewed again by the larger research team (JLS, CAJ, CJ, RM, and GDC) for further input and discussion, resulting in finalized summaries. While inter-coder reliability was not formally assessed, the qualitative team (JLS, CAJ) met weekly during the interviewing and analysis process to aid in the transparency and consistency of interview protocols, template coding, and creation of topical summarization reports. Our content analysis process also allowed for multiple, iterative steps along the way, including feedback from the larger research team, to aid in the trustworthiness of our interpretations. Additionally, the Consolidated Criteria for Reporting Qualitative Research (COREQ) was employed to guide rigor in the presentation of our findings [18].

### Results

We completed 20 interviews, 15 with patients and 5 with providers. Of the 15 patients, 11 completed the blood test, and 4 agreed to the blood test but had not completed it (Table 1). Those who completed the blood test had an average age of 58 years (range: 51–73), were mostly female (73%), and had been a member of the health system for an average of 21 years (range: 5–56). Those who did not complete the blood test had an average age of 64 years (range: 52–76), were mostly female (75%), and had been a member of the health system for an average of 20 years (range: 7–32). The 5 provider interviews included 3 primary care providers, one gastroenterologist (GI), and one medical assistant (MA) in the GI department that schedules follow-up colonoscopies following abnormal CRC screening results (e.g., FIT).

**Table 1. Patient description.**

| Characteristics | Completers* (11) | Non-Completers (4) | Totals (15) |
|---|---|---|---|
| | N | N | **N** / (%) |
| **Gender** | | | |
| *Female* | 8 | 3 | **11** (73%) |
| *Male* | 3 | 1 | **4** (27%) |
| **Average Age (range)** | | | |
| *51–55* | 5 | 2 | **7** (47%) |
| *56–76* | 6 | 2 | **8** (53%) |
| **Race / Ethnicity** | | | |
| *White, non-Hispanic* | 9 | 4 | **13** (87%) |
| *Hispanic* | 1 | - | **1** (6%) |
| *Black* | 1 | - | **1** (6%) |
| **Average Membership Years at Health System** | | | |
| *5–15* | 6 | 1 | **7** (47%) |
| *16+* | 5 | 2 | **7** (47%) |
| *Unknown* | 0 | 1 | **1** (6%) |
| **Any Prior CRC Screening History in Past** | | | |
| *No* | 6 | 2 | **8** (53%) |
| *Prior FIT Screening Ever* | 3 | 0 | **3** (20%) |
| *Prior Colonoscopy Ever* | 2 | 1 | **3** (20%) |
| *Unknown* | 0 | 1 | **1** (6%) |

*All completers had a normal blood test resulted; CRC = Colorectal cancer; FIT = Fecal Immunochemical Test

## Patient experience and feedback

**Prior CRC screening and reasons for not completing a FIT in the past.** Regarding prior CRC screening history, six patients (40%) self-reported screening in past, with three reporting past colonoscopy due to symptoms rather than an abnormal FIT, and three reporting a past FIT (two of these stated the FIT was never processed). Nearly all patients (10/15) cited forgetfulness or procrastination as a reason for not completing a recent FIT. Some patients said the FIT was "*out of sight and out of mind*" and others that it was "*put on the pile of things to do later*" and then never done. Most patients (10/15) also noted that the test was "*not as simple as it seemed*" as completion required several steps, such as placement of FIT in bathroom, reading instructions, timing completion to when having bowel movement, labelling completed sample properly, and mailing completed kit, that contributed to their lack of follow through. As one patient described, "*It seemed time consuming to do the [FIT] test. It was like I don't want to do that right now I'll do it later and I would just forget. And it was just not something that was simple—I am like I don't want to do that right now. Or the test would expire so I'd have to get another one. . .So it wasn't like I was defiant. I have to be motivated sometimes to do something, so it wasn't motivating enough to do it*" (completer, female, 10912).

Additionally, some patients (7/15) reported experiencing health issues, such as runny stool, that made FIT completion difficult, while other patients (7/15) assigned testing a low priority or dismissed it. Finally, fewer than half of the patients (6/15) reported experiencing disgust or an 'ick' response in dealing with their own fecal matter. One patient noted: "*I had been delaying doing it [the FIT test] because of the gross factor*" (completer, female, 10987).

**Reasons and motivations to choose blood-test option.** All patients, even those who eventually did not complete it, reported excitement and a sense of relief about being offered a

blood test. One third of patients (5/15) noted they had no intention of ever completing the FIT test, so were grateful for the blood test opportunity. One patient stated: "*I screamed hallelujah. I mean I really did. That would be so awesome!*" *(non-completer, female, 10712)*. Despite the enthusiasm, this patient did not end up completing the test because her clinical appointment, for which the blood draw was tied to, was canceled. Overall, patient excitement generally stemmed from not having to do a test that seemed "*kind of gross*" or relief from guilt they experienced for not having previously completed the FIT test.

Patients often noted multiple reasons for wanting to complete or completing the blood test. All patients perceived the blood test as simple and more convenient, and a task that could easily be added to a doctors' visit or to a blood draw done for another reason. As one patient described, "*I just thought it'd be easier. I've had so many blood tests that I got very used to having to go up to [location], and so it was like nothing—I knew the parking arrangement and the best time to go and everything so it was very easy*" *(completer, male, 10531)*. Patients also described frequently (13/15) how they would be more likely to be accountable to following through on a blood-draw appointment, than completing a FIT at home. One patient described: "*I actually liked the idea because it was a specific time and place. I had to show up, I had to do the [blood] test. I couldn't just put it off forever, which is what I do with the FIT test. So that part of it was very positive for me*" *(completer, female, 10764)*. Another patient stated: "*The blood draw would be a quick and easy way to get tested and I was eager to complete the screening when I had another appointment with my provider scheduled*" *(non-completer, male, 10794)*. Notably, many patients (12/15) also perceived the test to be more accurate than the FIT. "*I don't know how true it is, but I just felt like this way [blood test] they would be able to tell if there's anything in my system through the blood*" *(completer, female, 10518)*. Additionally, about two-thirds of patients (10/15) appreciated not having to "mess" with the multiple-step process of the FIT (instructions, timing of collection, labelling FIT, remembering to mail it). As a patient noted, "*I would rather have a blood test and do it real quick, so I don't have to do the icky parts [of FIT test]—the collecting of it, and taking the samples and putting it on the card and all that kind of stuff—you know and then mail it off.*" *(completer, female 10943)*.

**Questions and concerns.** The majority of patients (12/15) had no questions or major concerns about the blood-test option; three patients reported wanting to better understand the accuracy of the blood test and how it differs from the FIT. One patient stated, "*The only question I had was how accurate it [blood test] was compared to the other test [FIT]. If I would've done the other test [FIT], would the results be the same?*" *(completer, female, 10912)*. Another noted, "*What's the difference between the blood test and the FIT test, I just don't know that much about it*" *(non-completer, female, 10449)*.

**Reasons for not completing blood test.** For the four patients that scheduled but did not complete the blood draw, all stated that the reasons were logistical rather than because of concerns or fears about the blood test. These patients often shared more than one barrier to completion, including: travel concerns due to a winter storm (2/4), general transportation challenges (2/4), competing family priorities (1/4), issues with coordinating the blood draw with the primary care visit (1/4), and not feeling well (1/4). One participant described, "*It was because of the winter storm, we were stuck. . .also I really wanted to be able to do it while I was going to get my blood draws anyway. And they said that wasn't possible. And that was my biggest reason for not doing it*" *(non-completer, female, 10991)*. Another participant shared, "*I don't think I ever made it over there because it was on the same day as an appointment with my son. And that sort of overtook my day*" *(non-completer, male, 10449)*.

**Reaction to results.** For the 11 completers of the blood test, nine recalled receiving the result letter in the mail, and all found the wait-time for results (several weeks) acceptable. Eight of the nine patients found the result easy to understand, while one patient was initially

confused by the lay-out of the letter. This patient reported, "*I completely missed the big letter that said negative. And then I looked at it again and realized that that was up at the top of the page in very bold letters.*" *(completer, female, 10764)*. All 9 of these patients reported feeling comfortable and confident in their blood-test results. For the two patients without results at the time of the interview, both noted not checking their mail frequently and expecting the result to be posted to their online health portal, similar to how they experience other lab results reported. None of the 11 patients had had opportunity to discuss the results with their providers yet, but four patients reported some confusion as to why they were still being prompted to complete a FIT test even though their blood test was normal. As one patient noted, "*I got [blood-test results] via mail, all very clear. . .but the [FIT] still keeps popping up on my healthcare reminder that I need to do my little [FIT] test—so how to tie the two together. . .?*" *(completer, female, 10954)*

**Future actions.**   Overall, and if given a choice, most patients (14/15) who had passively declined FIT stated that they would choose the blood test over FIT because it was simpler, easier, and more convenient than FIT testing. One patient described, "*I would 100% do this blood test [if given a choice]. . .it was really convenient but even if it wasn't I would still make this choice even if I had to make a special trip to the lab to have blood taken.*" (completer, female, 10987). The one patient who was more neutral desired choice in blood-draw location to help facilitate completion in any future offering of the blood test. Additionally, many patients (14/15) noted that following through with a blood-draw appointment or combining a blood draw with a provider visit was logistically easier than completing an at-home FIT with its multiple steps (e.g. remembering, completing, and mailing). One patient described, "*I feel like it was just as trustworthy as all the other tests that we take. . . I don't like the yearly FIT. I don't like how you have to get your stool. . .this way [blood test] it's clean. Fast. And I don't have to carry it around.*" (completer, female, 10518).

The once-every-three-year testing interval was considered easy and acceptable by all patients. One patient stated: "*I love that. . . I would much rather do it every 3 years if it was just a blood test, than every year*" (completer, female, 10943). When asked about willingness to pay for the blood test, all patients stated a $25–40 co-pay would be acceptable and consistent with other laboratory fees. A patient summarized, "*I'm willing to pay the standard co-pay*" (non-completer, male, 10449). A few patients (4/15) expressed willingness to pay even more for the convenience and ability to complete CRC screening. "*It's almost like my contact lenses at $500. . .I would still pay for it [blood test]. I would figure out a way to do it. That is how important it is to me*" (completer, female, 10518).

## Provider reaction and feedback

**Awareness and general reaction.**   Three of the providers had no prior awareness of blood-based CRC screening tests prior to our interview and sharing of information about the Guardant Shield test. Two providers, one GI and one PCP, were familiar with the blood-test options. Overall, all providers interviewed expressed general optimism about the idea of an alternative to stool testing or colonoscopy. As one provider summarized, "*I've definitely heard of these tests and the potential availability for them. . .I think it will be a greater acceptance for doing blood-based tests over having to do FIT tests*" (PCP, male, 105).

**Advantages of blood test.**   All providers interviewed recognized that blood tests potentially hold many advantages over stool testing. Providers recognized patients are often non-compliant with FIT because it requires multiple steps to complete, deals with fecal matter, and is perceived to have no "deadline". A provider stated, "*So it's just like all these steps [in doing a FIT]. . .What I really like about this blood test is just the practical nature, once the patient buys*

*in and says, 'yeah I'll do it', you just go to the lab and you get it done...' (PCP, female, 104).* Providers noted ordering a blood draw is an easy, efficient, and familiar process for patients, that patients could easily tie the blood draw to other lab or in-person visits, and in general, patients are very compliant with completing blood draw orders. One provider described, "*It's very easy as a clinician to say, 'okay we need to just check your routine labs' and people are very willing to do that at the time... I have a suspicion that if a blood-based test becomes available it will be more popular and a higher percentage of people will be interested in doing it over the stool check...' (PCP, male, 105).* Another provider stated, "*Definitely, stool is a very inconvenient thing for patients to do it. It's just not going to work. Blood test is something that would get a lot of compliance" (GI, male, 101).* Finally, all providers also appreciated the blood test would ease the burden on both providers and patients to have to constantly remind regarding yearly FIT testing, and that the 3-year screening interval of the blood test would be acceptable to patients.

**Disadvantages of blood test.** Concerns about the blood test centered on cost issues, whether patients would be worried about a positive test even if follow-up colonoscopy was negative, and whether the blood test could accurately detect precancerous polyps. Three providers (all PCPs) expressed concern about the possible total out of pocket cost of the blood test (estimated at $400-$600), and even the expected co-pay of the blood test (estimated at $25-$40) creating cost barriers and choice inequities for patients. A provider described, "*I would say in primary care if I had to guess, that for 10–15% of patients cost, even if it seemed pretty negligible, was an issue." (PCP, female, 103).* Another three providers (2 PCP, 1 GI) were concerned patients would not understand what a "DNA" blood-based test means, especially if the follow-up colonoscopy showed nothing concerning. These providers noted this may create patient worry, confusion or even requests for additional testing. One PCP said, "*The only thing that would probably bring up more unintended consequences would be additional screening based on getting a positive [blood] test and having a negative colonoscopy. Because then somebody would think logically that 'well gee if I have some kind of DNA saying that there's a tumor in me, where is the tumor*?' *So that may entail additional scans or things like that." (PCP, male, 105).* Finally, while most providers (4) had no concerns regarding the accuracy of the blood test, the GI provider did express concern regarding the accuracy of the blood test and its ability to identify precancerous polyps, stating, "*I think we need to be able to have a blood test that [can] detect a fair amount of advanced adenomas and that we can provide our patients and society the confidence that we're truly catching it before it becomes a cancer" (Male, 101).* The other four providers indicated no concerns regarding accuracy of the blood test and were more concerned with the sensitivity of the blood test being on par with the current FIT offering. One PCP stated, "*It [FIT] looks like it's anywhere between 60–80% for sensitivity so it looks like this [blood test] is a better test for sensitivity. That's actually reassuring." (PCP, female, 104).*

**Experience communicating results.** The MA scheduler and GI provider involved in communicating abnormal blood-test results and scheduling the follow-up colonoscopy reported experiencing both successes and challenges in communication efforts. Most patients reached for abnormal results were scheduled for a colonoscopy even if they were initially resistant or unsure. The MA scheduler stated, "*By and large we got most patients scheduled...once we kind of communicated their reason as to why...*" (Female, 102).* The development of talking points to answer patients' questions or concerns, along with clear documentation phrases for use within the EHR to communicate results with both providers and patients were also perceived as helpful in the result communication process: "*And then [lead GI provider] created a blurb of what the expectation was*: *'If negative then no further follow up. If positive a screening colonoscopy is recommended'...I added [documentation]], so I feel like it was pretty well laid out for all the providers to know that this is being sent as a result of the positive Guardant test" (MA Scheduler, female, 102).*

The main challenges in the results communication process centered on patients' being surprised or confused by an abnormal result and hesitant about the need for a follow-up colonoscopy. However, the MA scheduler noted the initial resistance to the follow-up colonoscopy was similar to what they hear post abnormal FIT, with patients expressing fear of colonoscopy, disbelief of screening results, and/or prioritizing other health issues. The MA scheduler said "*I would say over 50% of patients pushed back on the seriousness or the need for it [follow-up colonoscopy]. I think it's common. I don't think better or less [than abnormal FIT], I think it's about average. . .*" *(Female, 102)*. Additionally, the GI provider also noted some patients in the study did not seem to understand an abnormal blood-test result would mean a referral for a colonoscopy, stating, *"One of the biggest concerns is patients just don't seem to have understood the concern [if blood test abnormal]." (Male, 101)*.

**Future actions.** Most of the providers (3 PCPS, 1 MA) were very open to having a blood test as a choice for patients in the future. Providers felt it would be easy to implement and incorporate into the current in-clinic and population-based mail out approaches to CRC screening. One provider stated, "*If there was an order, someone wouldn't even necessarily have to make an extra trip into the lab. (PCP, female, 103)*. Furthermore, incorporating the blood test as part of the CRC screening offerings was viewed as something that could improve both quality goals for the organization as well as patient compliance, as noted by one provider: *I think it would be a game changer for our CRC screening [rates] and HEDIS measures. (MA Scheduler, female, 102*). However, for one provider (GI), improvements in the accuracy of the blood test would be needed before implementing the blood test as a screening option.

## Patient and provider advice

Both patients and providers observed "many upsides" to offering a blood-based CRC screening option (Table 2). Both groups desired the blood test as a CRC screening option to everyone due for screening, not just FIT noncompliant patients. Both wanted blood-draw options at any clinic location and/or tied to other visits or blood draw appointments. Both desired educational materials that simply explain what the test is for, how it differs from FIT and colonoscopy, differences in accuracy, intervals, and costs across the screening options, what a blood-based "DNA" type of test means, and the importance of follow-up colonoscopy if abnormal. As one patient stated, *"I think anybody would benefit from a conversation about accuracy. If it could be combined with other blood draws, that would be ideal. And having it available at all the locations" (completer, female, 10764)*. Providers also suggested a multi-pronged educational roll-out campaign geared toward informing all providers about the blood-test option delivered both via presentation and by email that incorporated fact sheets or talking points to use with patients. A provider suggested, "*It would be nice to have from a primary care provider education standpoint and [for the] patient, a visual [showing] this is the efficacy of FIT testing for advanced adenoma versus colon cancer. This is the blood-test, and this is a screening colonoscopy." (PCP, female, 103)*. Additionally, providers and patients recommended development of EHR tools that aid both in accurately tracking and reminding regarding CRC screening (e.g. correct screening interval based on last completed option).

## Discussion

Our qualitative study yielded valuable insights into patients' and providers' perceptions about usability and acceptability of a blood-based screening test for CRC. Interviews showed that most patients who did not complete a FIT as part of an organized FIT screening program were receptive to completing a blood draw to undergo a commercially available blood test for CRC screening. Patients, whether they completed the blood draw or not, cited simplicity, ease, and

**Table 2. Patient and provider future recommendations.**

| | Patient Suggested | Staff Suggested |
|---|---|---|
| **Offer blood test to all due for CRC screening** | | |
| • Have blood-test option be a choice among all CRC screening options, not just an option for FIT non-compliant patients | √ | √ |
| • Incorporate blood-test option as part of overall CRC population-based screening programs | √ | √ |
| • Allow blood draw for screening test to be completed at any location of choice for patient and with ability to tie it to other visits or blood draw appointments | √ | √ |
| • Assure equity in cost of blood-test option | √ | √ |
| **Develop educational materials, tools, and training** | | |
| • Written and visual explanations about the blood test for patients | √ | √ |
| • Talking points for providers to aid in explaining blood test | | √ |
| • Educational materials for patients and providers should include following: *• How test differs from FIT and colonoscopy, including accuracy, interval, and cost • Lay language explaining what is meant by a "DNA" type of test • Emphasis on importance of follow-up colonoscopy if blood test abnormal • Lay language explaining how to understand a "normal" colonoscopy after an abnormal blood test* | √ | √ |
| • Training sessions for providers utilizing multiple delivery modes to inform of blood-test option and any related tools, such as: presentations at staff meetings, huddles, and repeated emails | | √ |
| **Develop tools in Electronic Medical Record (EMR) for tracking and documentation** | | |
| • Ensure patient's last CRC screening is easily identifiable by PCP in EMR, such as on a viewable "problem list" | √ | √ |
| • Ensure any EMR "flags" that track last CRC screening are appropriately turning "on and off" according to type of screening completed and its interval (e.g. FIT completion = every 1 year or if blood test completion = every 3 years) | | √ |
| • Develop smart phrases and automated messaging in EMR for providers to use in populating referrals to gastroenterologists or patient educational materials | | √ |
| • Ensure blood-test results are easily findable by both patients and providers via the EMR portal or patient APP | √ | √ |
| • Create a "batch" ordering process for the blood test that allows automated ordering of blood test on behalf of provider and a reminder prompt (in EMR portal or APP) for the patient to receive message their blood test is due | √ | √ |

CRC = Colorectal cancer; FIT = Fecal Immunochemical Test

convenience of the blood test. The blood test was viewed as an option that could help overcome barriers to completing FIT testing such as patients' forgetfulness or procrastination, the feeling that FIT is complicated to do, or feelings of disgust or an "ick" response. Reasons for not completing the blood test for CRC screening were logistical and not related to fear or hesitancy of the blood draw. Providers also favored blood testing as an alternative to other screening modalities but wanted assurance that the test would be affordable and could be offered to all screening-eligible patients. Providers also desired clear guidance on how to explain a normal colonoscopy result following abnormal blood-test result. Our study adds to limited information about providers' and patients' receptivity to blood-based testing for CRC screening. Because patient and provider perceptions can affect test use, our findings can guide possible future implementation of blood-based tests in care delivery settings.

Davis and colleagues' scoping review noted factors favoring blood-based testing were its non-invasiveness and perceived safety, convenience, and effectiveness [4]. Our findings were mostly consistent with this review. In contrast to these findings, our patients did not mention

non-invasiveness or perceived safety, a finding that was likely due to our standard of care being FIT testing, rather than more invasive colonoscopy screening. It is notable that patients' perception of test accuracy was not based on the reported test characteristics, as these were not shared with patients. Instead, patient participants felt that 'it would make sense' that the blood test would be more accurate than a small sample of stool in detecting CRC. Notably, interviewed patients reported being willing to pay standard co-pays for the blood-based test and a minority of patients reported a willingness to pay even more. Providers also expressed concerns about inequities that could be created if the blood test cost was substantially higher than other screening options.

The Centers for Medicare & Medicaid have recently issued a national coverage determination for blood tests approved by the US Food and Drug Administration. These tests must report a sensitivity of at least 74% and a specificity of at least 90% for cancer detection [8]. The current test's performance aligns with these predefined criteria. It's worth noting that the determination didn't establish performance thresholds for detecting advanced adenomas. As a result, this coverage determination is expected to promote widespread adoption of this or similar blood tests for CRC screening, pending approval, even though their performance is inferior to other available tests for advanced adenoma detection ((FIT 23–28% sensitivity, 94–96% specificity) and FIT-DNA (43% sensitivity; 89% specificity)) [1].

Our findings can influence possible future implementation of blood-testing approaches to CRC screening in primary care, including practical suggestions for patient and provider education and development of tracking and documentation tools in the EHR. We included PCPs and gastroenterologists in our study because in Kaiser Permanente's system, the vast majority of providers recommending CRC screening fall into these two categories of provider. While it is theoretically possible other categories of providers could refer a patient for CRC screening, given the lack of data on how any providers view a blood based test, or would use information about blood-based testing to encourage resistant screeners, we thought it was important to focus initial qualitative research efforts on these two groups of providers.

Notably, the providers we interviewed were interested in making the test available to all eligible patients, which contrasts with use as a second-line screening option (for those who decline other tests) [19, 20]. Concerns were also expressed about what to communicate to patients who have an abnormal blood test result yet have a normal colonoscopy result. The GI medical assistant also noted that s/he had to convince some patients to schedule a follow-up colonoscopy after an abnormal blood-test result, although expressed barriers were perceived as similar to those for follow-up colonoscopy after an abnormal FIT. Our findings may also influence strategies to more effectively promote uptake of FIT: patients' desire to have a schedule to complete the FIT could be achieved using 'poop on demand' or '"Go Before You Go" strategies that encourage patients to complete the test while at the clinic [21]. Alternatively, mailed outreach that is timed to upcoming appointments could facilitate FIT completion and return.

## Limitations

Our study has several limitations. We limited our recruitment to patients who had passively declined FIT testing, raising the possibility that our respondents had more favorable perceptions of blood-based testing and/or experienced more barriers to FIT testing than the general population. By pairing the blood test offer with an upcoming clinic visit, we attempted to mimic how a patient would be offered blood testing during standard clinic encounters. Our study also introduced obstacles that would not be present if blood testing were offered as standard care such as research-specific steps (e.g., signing a study consent form, scheduling a

separate appointment for the blood draw). Also, patients could only undergo the blood draw during the 4.5-month recruitment phase. Sample sizes were small, and we selected providers using a snow-ball approach and who did not necessarily have patients in the study; thus, they were likely not representative of all provider perceptions. Additionally, only one provider was a GI specialist and his views on the blood-test option were more cautious. Future research needs to explore differences in acceptability of the blood test across provider types (primary care versus specialty). We interviewed only fifteen patients, four of whom were non-completers of the blood test, so may not have obtained a full range of patient reasons for completing or not completing the blood test. Nevertheless, we observed consistency in the content of responses across both providers and patients [22]. We were not able to assess income of participants nor educational background, but these social-economic factors may be relevant to perceptions of blood-based testing and should be examined in future studies.

Additionally, as researchers who have studied uptake and implementation of CRC screening for decades, we are aware that there is a subset of people who have strong feelings about stool testing, and/or colonoscopy, and that adding to the arsenal of screening methods could play an important role in raising screening rates in eligible patients. As such, our concern was primarily to assess acceptability and comprehension of the pros and cons of blood-based screening, as a precursor to future and ongoing studies assessing performance of blood-based testing per se. Since blood-based CRC screening is more expensive than stool testing, it is important to try to narrow the subset of patients in whom this form of screening could be essential in preventing CRC. However, our preconception that raising screening rates for CRC is a desirable goal may have influenced the nature of questions asked during the interview, or even the participation of some providers or patients.

## Conclusion

Given the dearth of prior literature on patient and provider perceptions on CRC blood-based screening, our findings make a substantial contribution to the literature. All 15 patients were receptive to completing a blood draw for a commercially available blood test for CRC screening citing simplicity, ease, and convenience of the test–with 11 completing it and 4 not completing it due to logistical barriers. Providers also were pleased with ease of blood testing but wanted assurance that the test would be affordable, and desired guidance on how to best explain a normal colonoscopy following an abnormal blood test result. Future implementation of blood-based tests in care delivery settings may require patient and provider education and effective EHR-based documentation tools.

## Supporting information

**S1 Checklist. COREQ (COnsolidated criteria for REporting Qualitative research) checklist.**
(PDF)

**S2 Checklist. Human participants research checklist.**
(DOCX)

**S1 File. Interview guides.**
(DOCX)

**S2 File. Provider snapshot.**
(DOCX)

**S3 File. Code template examples.**
(DOCX)

## Author Contributions

**Conceptualization:** Jennifer L. Schneider, Gloria D. Coronado.

**Data curation:** Jennifer L. Schneider, Cheryl A. Johnson.

**Formal analysis:** Jennifer L. Schneider, Cheryl A. Johnson.

**Funding acquisition:** Gloria D. Coronado.

**Investigation:** Jennifer L. Schneider, Rajasekhara Mummadi, Gloria D. Coronado.

**Methodology:** Jennifer L. Schneider, Gloria D. Coronado.

**Project administration:** Charisma Jenkins.

**Writing – original draft:** Jennifer L. Schneider, Gloria D. Coronado.

**Writing – review & editing:** Jennifer L. Schneider, Cheryl A. Johnson, Rajasekhara Mummadi, Gloria D. Coronado.

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
