## [Decision Letter · Decision Letter 0]

31 Aug 2023

PONE-D-23-20071“I was screaming hallelujah”: Patient and provider perceptions of blood-based testing for colorectal cancer screeningPLOS ONE

Dear Dr. Coronado,

Thank you for submitting your manuscript to PLOS ONE. After careful consideration, we feel that it has merit but does not fully meet PLOS ONE’s publication criteria as it currently stands. Therefore, we invite you to submit a revised version of the manuscript that addresses the points raised during the review process.

Please note that we have only been able to secure a single reviewer to assess your manuscript. We are issuing a decision on your manuscript at this point to prevent further delays in the evaluation of your manuscript. Please be aware that the editor who handles your revised manuscript might find it necessary to invite additional reviewers to assess this work once the revised manuscript is submitted. However, we will aim to proceed on the basis of this single review if possible. 

The reviewer has raised a number of concerns that need attention. They request additional information on methodological aspects of the study to comply with reporting standards for qualitative studies. They also request revisions to better discuss this manuscript in the context of the literature and the interpretation of the results. Please take care to ensure that your conclusions are not overstated. Could you please revise the manuscript to carefully address the concerns raised?

We look forward to receiving your revised manuscript.

Kind regards,

Marianne Clemence

Staff Editor

PLOS ONE

2. Thank you for your ethics statement, which states "Written informed consent was obtained for all participants who attended the laboratory visit to receive the commercially available blood test." Please could you clarify whether all participants gave their written informed consent to take part in the interviews, including any patients who did not attend a laboratory visit, and providers.

“I have read the journal's policy and the authors of this manuscript have the following competing interests: From 2020- 2022, Dr. Coronado served as a scientific advisor for Exact Sciences, through a contract with the Kaiser Permanente Center for Health Research.

5. Please update your submission to use the PLOS LaTeX template. The template and more information on our requirements for LaTeX submissions can be found at http://journals.plos.org/plosone/s/latex.

Reviewers' comments:

Reviewer's Responses to Questions

**Comments to the Author**

1. Is the manuscript technically sound, and do the data support the conclusions?

Reviewer #1: Yes

2. Has the statistical analysis been performed appropriately and rigorously? 

Reviewer #1: N/A

3. Have the authors made all data underlying the findings in their manuscript fully available?

Reviewer #1: No

4. Is the manuscript presented in an intelligible fashion and written in standard English?

Reviewer #1: Yes

5. Review Comments to the Author

Reviewer #1: PLOS ONE Review:

This qualitative study assesses perceptions of blood-based colorectal cancer screening modality among patients who did not previously complete/adhere to a stool-based screening test. Perceptions of several types of health providers are also assessed. This is a well-written manuscript and overall, the findings are interesting, and I believe have important implications for increasing screening and patient comfort/satisfaction with information about screening. A more robust background on the blood test is needed to orient the reader and several implications of the findings can be expanded on in the discussion. Details comments are below.

Introduction:

• Authors state that blood-based testing for cell-free DNA is the newest option. It would be helpful to know the general availability dates of FIT tests and blood tests for context. What year was blood-based approved? What type of USPTF rating does it have? Given that the authors note that the commercially available blood test that was used was not approved by the FDA at the time of the study, some additional rationale for the use of the blood-based test is needed. Also, is it currently FDA-approved, and on what date did this occur?

• Authors note that participants “passively declined” a FIT test. It is unclear what this exactly means. Perhaps more detail in the methods would help to understand the use of this term.

Methods:

• Some of the detailed information about the Shield blood test in the Methods section may be more appropriate in the background section. Perhaps add this as a comparison to the FIT test since this comparison is made as part of the rationale.

• The statement that patients were notified that the test is not validated for stand-alone use in diagnosis is a bit misleading without additional context. For example, does this mean that the full protocol for a positive on this test includes a follow-up test? If so, adding that context would clarify for the reader. Also, this is the same protocol for FIT (positives are followed up with a colonoscopy, so again, this may be information more suitable in the introduction section to set the reader up to understand the nuances of each type of test (The FIT, which was declined, and the subsequent blood-test).

• What input of the research team informed semi-structured guides? Also, how was the literature used to inform guides? Were items adapted from similar types of studies or were themes used to develop new questions? Also, was a code book used for coding and was the data managed with any specific tools? (Excel, Nvivo, other?) In general, I may be helpful to review a qualitative checklist to ensure all details are included (SRQR or COREQ checklists).

• Also, were interviews recorded? Transcribed? There is mention to showing a visual to providers during interviews, but not clear how that was conducted if interviews were conducted via phone, were visuals emailed to providers prior to the interview?

• What was the rationale for interviewing patients who did and did not complete the blood-based test? Were there specific accrual goals for these groups, and were there plans to compare responses? If only interested in those who did complete the blood tests, why did you interview those who did not complete them? (I see that later in the results it indicates that those 4 were scheduled to complete it, perhaps that can be clarified in the abstract and methods), The way they are discussed sets them up as different groups of participants, when maybe it's more accurate to think of them as being interviewed at an earlier stage in the process of screening. Do you know if they eventually went on to complete the screening via blood? That would then give a bit more confidence that they are a homogenous group (e.g., declined FIT but completed or scheduled to complete a blood-based test).

• In the description of the participants is there a way to provide geographic information? For example, are participants in urban vs. rural areas? This may impact perceptions of a screening modality that requires in-person access vs. one that can be completed remotely.

Results:

• One concern with the results is that authors report that most (14/15) participants said they would choose blood tests over FIT, however, since most of the participants were already people who had declined fit and completed blood tests. I would just be careful not to indicate that blood tests are the preferred modality overall, b/c this specific population was already possibly in a position to prefer a different modality since they had already declined a FIT. Maybe even in the discussion, looking into if there are certain groups who may prefer blood over stool and if there are implications for screening campaigns using audience segmentation. Etc.

• I thought it interesting that providers were concerned about cost, but patient participants did not list that as a concern. What was the income range of participants?

Discussion:

• I thought it was interesting that participants viewed blood as more accurate than stool. Could authors provide some comparison of accuracy between tests in the intro or discussion?

• Also, was interesting that having the blood test scheduled was a perceived benefit for participants. Have there been any studies where FIT tests were tied to appointments or “due dates”? what are the implications for increasing screening rates based on some of these identified findings?

6. PLOS authors have the option to publish the peer review history of their article (what does this mean?). If published, this will include your full peer review and any attached files.

Reviewer #1: No

---

## [Author Response · Author response to Decision Letter 0]

25 Sep 2023

PLOS ONE Response to Reviewers PONE-D-23-20071

(Page numbers referenced below pertain to manuscript version with tracked changes)

This qualitative study assesses perceptions of blood-based colorectal cancer screening modality among patients who did not previously complete/adhere to a stool-based screening test. Perceptions of several types of health providers are also assessed. This is a well-written manuscript and overall, the findings are interesting, and I believe have important implications for increasing screening and patient comfort/satisfaction with information about screening. A more robust background on the blood test is needed to orient the reader and several implications of the findings can be expanded on in the discussion. Details comments are below.

Introduction

1) Authors state that blood-based testing for cell-free DNA is the newest option. It would be helpful to know the general availability dates of FIT tests and blood tests for context. What year was blood-based approved? What type of USPTF rating does it have? Given that the authors note that the commercially available blood test that was used was not approved by the FDA at the time of the study, some additional rationale for the use of the blood-based test is needed. Also, is it currently FDA-approved, and on what date did this occur?

Response: We have updated the Introduction and Discussion to add/clarify the requested information, to the extent we can. Some of the information the reviewer requested is not yet available, which we have also clarified. This is now better explained in the Introduction on pages 3-4 and Discussion on page 14. Specifically, we explain that the Guardant Health SHIELD test has not received FDA approval (it is in process) and has not undergone evaluation for recommendation by the US Preventive Services Task Force. Its performance characteristics align with the predetermined benchmarks set by the Centers for Medicare and Medicaid. This and other blood tests for colorectal cancer screening are expected to receive FDA approval in the coming year.

2) Authors note that participants “passively declined” a FIT test. It is unclear what this exactly means. Perhaps more detail in the methods would help to understand the use of this term. 

Response: We have supplemented the Methods section with additional information, clarifying that the individuals interviewed were part of the group that should have received colorectal cancer screening. As per the standard practice within the health system, they were given a FIT test within the last 3 to 9 months. However, these participants never completed and/or returned the FIT test, classifying them as individuals who chose not to actively pursue the FIT test option (page 4). 

Methods

1) Some of the detailed information about the Shield blood test in the Methods section may be more appropriate in the background section. Perhaps add this as a comparison to the FIT test since this comparison is made as part of the rationale.

Response. We have supplemented the Introduction (background) section with additional information about the blood test. We have added performance characteristics of FIT, as the reviewer recommends (pages 3-4).

2) The statement that patients were notified that the test is not validated for stand-alone use in diagnosis is a bit misleading without additional context. For example, does this mean that the full protocol for a positive on this test includes a follow-up test? If so, adding that context would clarify for the reader. Also, this is the same protocol for FIT (positives are followed up with a colonoscopy, so again, this may be information more suitable in the introduction section to set the reader up to understand the nuances of each type of test (The FIT, which was declined, and the subsequent blood-test).

Response. We have added text to the Introduction to make it clearer that a follow-up colonoscopy is needed for blood test results that are abnormal, same as that protocol for abnormal FIT results (pages 3-4). 

3) What input of the research team informed semi-structured guides? Also, how was the literature used to inform guides? Were items adapted from similar types of studies or were themes used to develop new questions? Also, was a code book used for coding and was the data managed with any specific tools? (Excel, Nvivo, other?) In general, I may be helpful to review a qualitative checklist to ensure all details are included (SRQR or COREQ checklists).

Response: We have clarified our guide development and analysis process in the Methods section, and utilized the COREQ checklist (pages 5-6) as suggested. We further explain on page 5 how we generally reviewed the literature to aid in our interview guide development and primarily relied on our research goals and the expertise of the study team for question development. We did not adapt items to our interview questions based on findings from other studies. Given the overall small number of interviews and short timeline of the pilot project, we chose to not utilize a software program such as NVivo for coding, but rather developed a Word-based code template based on review of the transcripts. We then extracted sections of the transcripts into our code template using cut and paste functions. Data within the coded template formed a report that was iteratively reviewed multiple times to identify and refine themes. We have now added additional description to our analysis process in the Method section, clarifying our coding/analysis process on page 6. We have also applied the COREQ qualitative checklist as noted in our method section (also referenced). 

4) Also, were interviews recorded? Transcribed? There is mention to showing a visual to providers during interviews, but not clear how that was conducted if interviews were conducted via phone, were visuals emailed to providers prior to the interview?

Response: Thank you for the opportunity to clarify these methodology questions. In the Methods section under analysis, we note that we had already stated all interviews were recorded and transcribed (page 6). Under the data collection portion of Methods, we have now added clarifying language that the visual shared with providers as part of the interview discussion was emailed to those providers agreeing to the interview (page 5). 

5) What was the rationale for interviewing patients who did and did not complete the blood-based test? Were there specific accrual goals for these groups, and were there plans to compare responses? If only interested in those who did complete the blood tests, why did you interview those who did not complete them? (I see that later in the results it indicates that those 4 were scheduled to complete it, perhaps that can be clarified in the abstract and methods), The way they are discussed sets them up as different groups of participants, when maybe it's more accurate to think of them as being interviewed at an earlier stage in the process of screening. Do you know if they eventually went on to complete the screening via blood? That would then give a bit more confidence that they are a homogenous group (e.g., declined FIT but completed or scheduled to complete a blood-based test). 

Response: We have added text to our Methods on page 5 to better explain our rationale for interviewing patients who did and did not complete the blood-based test. We also made edits to emphasize that our goal was to explore any similarities and/or differences in motivations and concerns for completing the test. We apologize if it was not clear that interviewed patients are two distinct groups in terms of the blood-test completion outcome, but are a part of the same larger study on the same recruitment timeline (hence, one group was not interviewed at an earlier timepoint than the other). In the patient paragraph of Methods under data collection, we note we had already described how recruitment for both completers and those that no-showed to the blood draw occurred 4 to 8 weeks after their appointment (pages 5-6). In the third paragraph of the Methods section, we also explain that, given one of our main goals was to understand patient experiences of actually completing the test, we oversampled those patients who completed the blood test (pages 5-6). Furthermore, responses from the “completers” and “non-completers” of the blood test are compared throughout the presentation of our Result section along with illustrative quotes, including a separate section identifying reasons for non-completion. We also explain in our study setting section under Methods (page 4) that the blood test option was part of a larger study and not an aspect of standard practice within KPNW, hence patients that did not complete the blood test as part of our study would not have an opportunity to complete it at a later date. 

6) In the description of the participants is there a way to provide geographic information? For example, are participants in urban vs. rural areas? This may impact perceptions of a screening modality that requires in-person access vs. one that can be completed remotely. 

Response: We have added text to our Method section under study setting on page 4 to clarify that all participants (patients and providers) resided within an urban metropolitan area served by the health system. Additionally, we note in our Discussion the importance of understanding geographically, economically, and culturally/ethnically diverse perspectives of patients and providers in future research on acceptability of the blood test option (page 15).

Results

1) One concern with the results is that authors report that most (14/15) participants said they would choose blood tests over FIT, however, since most of the participants were already people who had declined fit and completed blood tests. I would just be careful not to indicate that blood tests are the preferred modality overall, b/c this specific population was already possibly in a position to prefer a different modality since they had already declined a FIT. Maybe even in the discussion, looking into if there are certain groups who may prefer blood over stool and if there are implications for screening campaigns using audience segmentation. Etc.

Response. The authors appreciate this comment. We have modified our Results and Discussion to suggest that patients who declined FIT testing were willing to undergo blood testing. Specifically, in the first paragraph of the Results, we state, “Overall, and if given a choice, most patients (14/15) who had passively declined FIT stated that they would choose the blood test over FIT because it was simpler, easier, and more convenient than FIT testing.” Moreover, in the first paragraph of the Discussion, we state, “Interviews showed that most patients who did not complete a FIT as part of an organized FIT screening program were receptive to completing a blood draw to undergo a commercially available blood test for CRC screening.” Notably, the limitations section already states this, “We limited our recruitment to patients who had passively declined FIT testing, raising the possibility that our respondents had more favorable perceptions of blood-based testing and/or experienced more barriers to FIT testing than the general population.” Because we did not interview individuals who had completed FIT, we could not appropriately comment on audience segmentation strategies. However, we have included ways to improve FIT testing programs based on our responses to Discussion 2) below.

2) I thought it interesting that providers were concerned about cost, but patient participants did not list that as a concern. What was the income range of participants? 

Response: While income of participants would be useful information for future studies, we did not collect the income range of participants as part of this pilot study. We discussed patients’ potential concerns about paying the estimated full price (400-600$), and included their sentiments that given the 3-year screening interval for the blood test, the estimated co-pay of $25-40 seemed reasonable. This finding is stated in our Results under “future actions” for patients (page 10). Additionally, as stated above in another response, we have now added in our Discussion the importance of future research on blood tests to obtain a diverse range of patient perspectives, including a range of income levels (page 15).

Discussion

1) I thought it was interesting that participants viewed blood as more accurate than stool. Could authors provide some comparison of accuracy between tests in the intro or discussion?

Response. The authors agree that this was a surprising finding. We provide a comparison of the accuracy of the blood test and FIT tests (sensitivity and specificity) in the Introduction (page 3). 

2) Also, was interesting that having the blood test scheduled was a perceived benefit for participants. Have there been any studies where FIT tests were tied to appointments or “due dates”? what are the implications for increasing screening rates based on some of these identified findings?

Response. We have updated the Discussion on page 15 to include a reference to previous evaluations of FIT screening programs that have promoted ‘poop on demand’ which allows patients to provide a stool sample during a standard clinic visit.

---

## [Decision Letter · Decision Letter 1]

28 Nov 2023

“I was screaming hallelujah”: Patient and provider perceptions of blood-based testing for colorectal cancer screening

PONE-D-23-20071R1

Dear Dr. Schneider,

We’re pleased to inform you that your manuscript has been judged scientifically suitable for publication and will be formally accepted for publication once it meets all outstanding technical requirements.

Kind regards,

Melissa J. Vilaro

Guest Editor

PLOS ONE

Additional Editor Comments (optional):

Please see and address the special points indicated by reviewer #2. 

Reviewers' comments:

Reviewer's Responses to Questions

**Comments to the Author**

1. If the authors have adequately addressed your comments raised in a previous round of review and you feel that this manuscript is now acceptable for publication, you may indicate that here to bypass the “Comments to the Author” section, enter your conflict of interest statement in the “Confidential to Editor” section, and submit your "Accept" recommendation.

Reviewer #2: All comments have been addressed

2. Is the manuscript technically sound, and do the data support the conclusions?

Reviewer #2: Yes

3. Has the statistical analysis been performed appropriately and rigorously? 

Reviewer #2: Yes

4. Have the authors made all data underlying the findings in their manuscript fully available?

Reviewer #2: Yes

5. Is the manuscript presented in an intelligible fashion and written in standard English?

Reviewer #2: Yes

6. Review Comments to the Author

Reviewer #2: The study provides valuable insights into patient and provider perceptions of blood-based colorectal cancer screening compared to stool-based tests. While the manuscript is well-crafted and the findings are intriguing, there are some aspects that need further clarification.

Specific Points:

1. Clarify the measures taken to ensure transcription accuracy, including any training provided to transcribers and the use of transcription software. Transcription quality is fundamental for maintaining data integrity.

2. Specify if inter-coder reliability was assessed. Multiple coders' consistency is vital for the validity of the analysis.

3. Detail the criteria used for determining relevant text for extraction to address concerns about data extraction consistency and comprehensiveness.

4. Elaborate on steps taken to ensure rigor and credibility, such as member checking or peer debriefing, following the COREQ guidelines. This would enhance the study's credibility.

5. Describe methods employed to mitigate reviewer bias during the identification of key content themes.

6. Discuss how researchers' backgrounds and perspectives influenced the study. Acknowledging reflexivity is crucial for transparency.

7. Confirm whether data saturation was achieved in the analysis section.

8. Specify the approach used for deriving themes to provide clarity on the analytical process.

Discussion/Limitations

1. Provide reasons behind recruiting patients who passively declined FIT testing and acknowledge potential bias. Clarify if active decliners were considered in the study.

2. Address the impact of the lack of diverse specialization among providers. Discuss how this imbalance might affect the study's findings and interpretations.

7. PLOS authors have the option to publish the peer review history of their article (what does this mean?). If published, this will include your full peer review and any attached files.

Reviewer #2: No

---

## [Editor Report · Acceptance letter]

13 Dec 2023

PONE-D-23-20071R1 

PLOS ONE

Dear Dr. Schneider, 

I'm pleased to inform you that your manuscript has been deemed suitable for publication in PLOS ONE. Congratulations! Your manuscript is now being handed over to our production team.

Kind regards, 

on behalf of

Dr. Melissa J. Vilaro 

Guest Editor

PLOS ONE